# Implementation fidelity of tuberculosis screening for Diabetes mellitus patients among healthcare providers offering diabetes services in Ubungo District, Dar es Salaam, Tanzania

Edwin Christian Chavala[1]*, Felistar Mwakasungura[2], Linda Simon Paulo[2], Tumaini Nyamhanga[2]

1 Department of Development Studies, Muhimbili University of Health and Allied Sciences, Dar es Salaam, Tanzania, 2 School of Public Health and Social Sciences, Muhimbili University of Health and Allied Sciences, Dar es Salaam, Tanzania

* edwinchavala5@gmail.com

## Abstract

Globally, the risk of acquiring Tuberculosis (TB) among Diabetes mellitus (DM) patients is three times higher than in the general population. Patients with DM not only have a high risk of getting TB disease but also have poor treatment outcomes. Despite the National TB guideline recommending TB screening among DM patients, adherence remains low. Therefore, this study aimed to assess the provider-level implementation fidelity (IF) and factors associated with TB screening among DM patients at public health facilities in Ubungo district. We conducted an analytical cross-sectional study from April 4th to May 25th, 2025, in 20 public facilities (3 hospitals, 5 health centers, 12 dispensaries) in the Ubungo district, using quantitative methods, among 94 health providers offering DM services. Data were collected through a questionnaire and analyzed for fidelity levels (low or high) using descriptive statistics. Then, a modified Poisson regression model using STATA version 16 identified factors associated with provider-level IF of TB screening among DM patients. The overall provider-level IF score was 83.0%, with (n=78) out of 94 providers self-reporting high fidelity to TB screening guideline components. Teamwork (aPR 2.28, 95% CI 1.11–7.12; p-value=0.032), self-efficacy (aPR 2.29, 95% CI 1.04–5.02; p=0.024), and facility-level the provider was working, especially at the hospital level (aPR 3.60, 95% CI 1.52–8.50; p=0.004) were significantly associated with provider-level implementation of TB screening for patients with DM. These findings suggest that collaborative teamwork, provider self-efficacy, and facility-level context influence the consistency of TB screening practices among patients with DM. Therefore, strengthening these factors could support improved implementation; however, further research is needed to establish effective strategies.

**Data availability statement:** The data supporting this study's findings are available on Figshare at the following link https://doi.org/10.6084/m9.figshare.31859437.

**Funding:** The authors received no specific funding for this work.

**Competing interests:** The authors have declared that no competing interests exist.

## Introduction

TB remains one of the deadliest infectious diseases worldwide, ranking as the second leading cause of death annually [1].According to the World Health Organization (WHO), the estimated global number of newly diagnosed TB cases reached 7.5 million in 2022, surpassing pre-pandemic levels [2]. Moreover, about one-quarter of the world's population is infected with Mycobacterium tuberculosis, with a lifetime risk of progressing to active disease ranging between 5% and 10% [3].People with weakened immune defenses or existing health conditions, such as DM, face a higher risk of developing TB [4]. There is also no doubt that DM has recently been recognized as a major risk factor for TB, ranking among the top five contributors to TB burden [5]. The sharp global rise in DM cases has highlighted the need to examine the intersection between DM and TB, positioning it as a key issue in public health [6]. Research indicates that individuals with DM are about two to three times more likely to develop TB compared with the general population, and this increasing trend of DM may complicate the TB prevention and control efforts [7]. More than half of the world's TB cases occur in countries that have significant prevalence rates and total numbers of DM cases [8]. Approximately 15% of patients with TB are found to have concurrent DM, a coexistence that has been identified as a potential catalyst for significant future public health challenges [9]. In response to this, the World Health Organization, International Diabetes Federation (IDF), and International Union Against Tuberculosis and Lung Disease recommended routine TB screening for patients with DM [10].

In Tanzania, despite the National TB guideline recommending TB screening among DM patients, adherence remains low, as it was found to range from 1.3–14%, which indicates that routine screening for TB among patients with DM is still often underdiagnosed [11]. The dual burden of TB among individuals with DM in Tanzania has not yet received sufficient attention as an emerging public health concern at both local and national levels, with a study carried out in Tanzania revealing that TB prevalence among people with DM was seven times greater than the national average [12]. Furthermore, findings in the Kagera region indicated that 14% of TB patients in Tanzania were also living with DM [13]. Not only that, but also, in a study that included Ubungo and other districts, it was revealed that out of 619 facilities reported to provide DM services, only 238 (38·4%) provided screening and treatment for TB [14]. Therefore, we conducted this study to assess the providers' level of IF and factors associated with TB screening among DM patients at public health facilities in Ubungo District.

## Materials and methods

### Ethics statement

The public health facilities involved granted permission. Ethical clearance (Ref No. DA.282/298/01. C/2735) was obtained on 21st March 2025 from the MUHAS ethical review committee. All participants were informed of the study aims, provided written consent, and were assured that participation was voluntary and that they had the right to withdraw at any time. No incentives were offered. Confidentiality was maintained throughout by using identity numbers instead of names.

## Study design and settings

An analytical cross-sectional study design employing a quantitative approach was used to collect and analyze data. The study was conducted in Ubungo District, located in Dar-es-Salaam Region. Ubungo is one of the five councils in the region, covering an area of 269.4 km². It shares borders with Kinondoni District and Kibaha (Pwani Region) to the north, Kisarawe (Pwani Region) to the west, and Ilala District to the south and east [15]. Among the five districts, Ubungo reported a noticeable rise in cases of DM throughout the two years, with a substantial rise of 2808 cases from 2022 towards 2024, representing a 13.4% increase, leading to a total of 23,538 confirmed DM patients based on data from the DHIS2 accessed at 12:23 hours on 15th January,2025 [16]. Even with these high numbers, there is limited research on how well TB screening for DM patients is being carried out by the DM healthcare providers in the Ubungo district.

## Study population

The study population included all clinicians aged 18 and above offering DM services in Ubungo district who were actively involved in direct patient consultations (i.e., personally seeing and screening) DM patients on at least the two scheduled DM clinic days per week in the past month, as they are the frontline staff responsible for daily screening of signs and symptoms, provision of patient education, counseling, and making referrals for all eligible patients attending public health facilities. Their direct role in patient care across Ubungo District allowed the study to accurately assess the extent to which TB screening practices are followed as per the National TB guideline among DM providers. The exclusion criterion was the healthcare providers who had been in their current DM service role for less than one month to ensure familiarity with routine practices and avoid including providers who may be unreliable or unrepresentative due to insufficient time to become familiar with routine workflows and guidelines in their current role.

**Sample size.** Cochran's formula (1977), $N = Z^2 p (1-p)/d^2$, was used to determine the minimum sample size required for the study [17]. According to this formula, n = desired sample size, z = critical value at 95% confidence level corresponding to 1.96, d = marginal error (desired level of precision, which is 5%), and p = prevalence, which in our study is the prevalence of adherence set at 13% [18] with a margin of error of 5%.

$$n = \frac{1.962 * 0.13 * (1-0.13)}{(0.05)2} = 174$$

To accommodate a 10% non-response rate, the sample size was adjusted as follows;

$$\text{Adjusted sample size} = \frac{174}{1-0.1} = 193$$

Since the total number of clinicians working among 20 public health facilities visited in Ubungo district was 168. Thus, the finite population correction for the proportion formula was used to calculate the sample size, as shown below [19].

$$n = no/ (1 + (no - 1)/N)$$

Where n = required sample size after finite population correction, no = Sample size assuming an infinite population, and N = Total population size

$$n = \frac{193/ (1 + (193-1)}{168} = 94$$

The minimum sample size of providers offering DM services required was 94.

**Sampling techniques.** The simple random techniques were used in selecting 20 public health facilities and 94 healthcare providers offering DM services in the Ubungo district. The proportional allocation formula was then used in determining the required sample per facility level [20]. This was worked out using the following approach: First, we determined the number of eligible providers per health facility level; Secondly, we employed the formula below to calculate representative sample size from each of the selected levels of health facility; Lastly, a simple random sampling technique was used to obtain the participants from each of the selected health facility levels.

$$ni = (n/N) Ni$$

Where: n = total sample size to be selected, N = total population, Ni = total population of each facility, and ni = sample size from each facility as shown in Table 1.

## Variables and measurements

Provider-level IF as the outcome of interest was measured using two constructs, which were details of content and frequency. Content was a Yes/No question (1 = yes activity done, 0 = No activity not done). At the same time, frequency referred to the extent to which clinicians at the facilities offering DM services performed TB screening activities on the guideline according to the prescribed frequency which was measured on a scale of 1–5 with a score ranging from 0 = never, 1 = rarely, 2 = for new clients 3 = For most clients 4 = All the time for all clients), with a score 4 being what the intervention guidelines expected [21]. We further re-grouped the original items selected from the two constructs and frequency into three domains, components which were TB signs and symptoms, TB awareness (education and

**Table 1. Participants' distribution per health facility.**

| Name of facility | Population per facility (Ni) | Sample per facility (ni) | Percent (%) |
|---|---|---|---|
| Sinza hospital | 35 | 20 | 21.3 |
| University of Dar-es-salaam (UDSM)- hospital | 16 | 8 | 8.5 |
| Ubungo district hospital | 9 | 5 | 5.3 |
| Goba Health Center | 6 | 3 | 3.2 |
| Makuburi Health Center | 14 | 8 | 8.5 |
| Mbezi health center | 9 | 5 | 5.3 |
| Makurumla Health Center | 22 | 12 | 12.8 |
| Kimara Health Center | 23 | 13 | 13.8 |
| Goba dispensary | 6 | 3 | 3.2 |
| Mburahati dispensary | 3 | 2 | 2.1 |
| Manzese Dispensary | 3 | 2 | 2.1 |
| Temboni dispensary | 3 | 2 | 2.1 |
| Mavurunza dispensary | 2 | 1 | 1.1 |
| Mpij magohe dispensary | 3 | 2 | 2.1 |
| Kibwegere Dispensary | 2 | 1 | 1.1 |
| Msumi dispensary | 2 | 1 | 1.1 |
| Msewe dispensary | 3 | 2 | 2.1 |
| Mabibo dispensary | 2 | 1 | 1.1 |
| Kinzudi dispensary | 2 | 1 | 1.1 |
| Amani dispensary | 3 | 2 | 2.1 |
| Total (N) | 168 | 94 (n) | 100 |

**Global Public Health**

counseling), and TB referrals based on our knowledge and experience. Each of the three domain components had different items that must be implemented to achieve the purpose of TB screening for DM patients among healthcare providers, as per the National TB guidelines [22]. For each of the three components, individual item scores were summed to generate a domain-specific fidelity score. To standardize the scores and facilitate interpretation, percentage scores were calculated by dividing each of the three component scores by its maximum possible value and multiplying by 100. Then the overall fidelity score was computed by merging the three domain components. Based on the various scores for all three domain components, the total maximum possible score was 40, with 16-items per provider. TB symptoms and signs had a maximum score of 25; TB awareness (education & counseling) had a maximum score of 10, and TB referrals had a maximum score of 5. To proceed with assessing the provider fidelity level, we conducted an internal consistency and reliability check to ensure that the various items are suitable to measure each of the three domain components. To evaluate the internal consistency of the grouped fidelity items within each domain component. The Cronbach's alpha statistic measure for consistency was calculated and showed acceptable to excellent internal consistency for the scale components [23]. TB signs and symptoms scored 0.70, TB education and counseling 0.97, while TB referral, assessed by a single item, did not require reliability testing. The factors associated with provider-level IF of TB screening among providers offering DM services were system factors such as (training, teamwork, records and documentation, staff allocation) while guideline related factors included (nature and source of the TB guideline, relative advantage and finally design quality and package of the TB guideline) which were measured based on a 5-point Likert scale, and were coded ranging from 1 for "strongly disagree" to 5 for "strongly agree" with 3 representing neutral position. Furthermore, provider factors such as the provider's familiarity with TB screening for DM patients, as per the National TB guideline, and providers' self-efficacy, which were measured and coded ranging from 1 for "very poor" to 5 for "very good". The table below describes the explanatory variables used in this study and their definitions as shown in Table 2.

## Data collection

A self-administered structured questionnaire (S1 Text) was used to collect data from participants. It was adopted and modified based on a study from Ghana that measured provider-level implementation fidelity of TB screening among healthcare providers [21]. The questionnaire gathered information on providers' demographic characteristics, their reported

**Table 2. Operational Definitions.**

| Variable | Description |
|---|---|
| Design quality and packaging | Perceived excellence in how the guideline itself is bundled, presented, and assembled. |
| Familiarity with TB guidelines | Refers to the extent to which healthcare providers know, understand, and are aware of the recommended procedures, criteria, and protocols for identifying TB in individuals with DM, and how easily they can access this information to guide screening in routine practice. |
| Fidelity | Refers to how well a specific intervention is implemented as intended, which is critically crucial in supporting effectiveness in particular research.<br>In our study, providers achieving ≥60% fidelity were classified as high-fidelity, and those with <60% as low-fidelity. |
| Guideline | A structured statement outlining systematic steps and synthesizing relevant information, grounded in scientific evidence, to support healthcare providers in following evidence-based practices. |
| Nature and source of TB guidelines | The perception of healthcare providers, whether knowing the guidelines, influences their implementation. |
| Relative advantage | Healthcare providers perceived a benefit or advantage of implementing the guideline. |
| Self-efficacy | Individual belief of healthcare providers in their own capabilities to execute courses of action to achieve implementation goals. |
| Teamwork | Refers to the way a team is organized and works together, including communication, trust, and shared goals to coordinate actions and successfully implement innovations such as clinical guidelines. |

adherence to clinical guidelines for TB screening among diabetes patients (covering content and frequency, as outlined in the TB/DM collaborative protocol in the National TB guidelines), as well as factors associated with provider-level IF, based on the Conceptual Framework for Implementation Fidelity (CFIF) [24]. Research assistants received three days of training, followed by a fourth day dedicated to pretesting. The pretest was conducted in a Mwananyamala hospital, Buguruni health center, and Bungoni dispensary, the facilities which had similar characteristics to the study sites but were not part of the main study. This process was done to minimize measurement errors, reduce respondent burden, and identify any problem areas in the tools for correction [25]. Data for this study were collected between April 4 th to 25th May 2025, in 20 public health facilities that provide DM services, including 3 hospitals, 5 health centers, and 12 dispensaries across Ubungo District in Dar-es-Salaam Region.

## Data management and analysis

The Epi Info version 3.5.1 software was used to capture data from a hard-copy questionnaire, which was subsequently imported into Stata 16. To ensure data security, the file was saved in a password-protected location accessible only to the investigator. Before analysis, a thorough cleaning process was undertaken to remove missing values, outliers, and other irregularities, and this involved verifying the accuracy and completeness of the information collected from participants. To assess the provider- level IF of TB screening for DM patients, normality tests using skewness were first performed using STATA version 16 (Stata Corp, College Station, TX, USA) to determine data distribution. Summary statistics were then calculated for IF using the two constructs of details of content and frequency, which were further merged into three domains: TB signs and symptoms, TB awareness (education and counseling), and TB referrals to get scores among healthcare providers. Provide -level IF was categorized as either high (≥60%) or low (<60%), based on a threshold adopted from a prior study in Ethiopia [26]. To examine factors associated with provider-level IF of TB screening for DM patients among healthcare providers, unadjusted and adjusted modified Poisson regression models were used [27]. These models evaluated the relationship between fidelity and various system, provider, and guideline-related factors. This started with Model 1, which assessed crude associations to determine individual association with fidelity to the National TB guideline, while Model 2 adjusted for potential confounders. Variables with an association that had a p-value < 0.2 were thereafter subjected to a multiple regression analysis to determine an independent association after controlling for confounders and covariates. Statistical significance was set at p < 0.05 [28]. The variance inflation factor (VIF) was computed to assess the presence of multicollinearity among the regression variables. For TB screening among DM patients, VIF values ranged from 1.08 to 1.47, with a mean VIF = 1.23, indicating no evidence of multicollinearity. All analyses were conducted using Stata version 16 [29].

## Results

### Demographic information of study participants

A total of 94 healthcare providers participated in this study, as presented in Table 3. Most of the providers offering DM services were aged 18–34 years, 61 (64.9%), and slightly more than half were male, 50 (53.2). The majority, 54 (57.5%), had a diploma or below, and worked primarily at dispensaries, 20 (21.3%), health centers, 41 (43.6%). Most providers had one to five years of work experience 71 (75.5%). The mean age of the study participants was 37 (±SD = 6.6) years.

### Descriptive statistics on the provider-level implementation fidelity measured through self-reported adherence to TB screening guideline components according to the National TB guideline

The following describes descriptive statistics for the raw data on provider-level IF for TB screening among providers offering DM services, as shown in Table 4. For example, 72.3% (n = 68) of providers reported that they provide TB education to patients with DM, while 93.6% (n = 88) reported referring patients with presumptive TB.

**Table 3. Social demographic characteristics of study participants (n = 94).**

| Characteristic | Category | n (%) |
| --- | --- | --- |
| **Age** | 18-34 | 61 (64.9) |
| | 35-60 | 33 (35.1) |
| **Sex** | Male | 50 (53.2) |
| | Female | 44 (46.8) |
| **Education level** | Degree and above | 40 (42.6) |
| | Diploma and below | 54 (57.4) |
| **Level of facility** | Dispensary | 20 (21.3) |
| | Health Center | 41 (43.6) |
| | Hospital | 33 (35.1) |
| **Years of working** | 1 to 5 years | 71 (75.5) |
| | Above 5 years | 23 (24.5) |

**Table 4. Provider-level implementation fidelity on the adherence to TB screening among DM patients.**

| Items | N = 94 | | | | |
| --- | --- | --- | --- | --- | --- |
| | No n (%) | Yes n (%) | | | |
| 1. Asks DM patients about cough for two weeks. | 58 (61.7) | 36 (38.3) | | | |
| 2. Asks DM patients about night sweats. | 52 (55.3) | 42 (44.7) | | | |
| 3. Asks DM patients about fever. | 5 (5.3) | 89 (94.7) | | | |
| 4. Asks DM patients about chest pain and coughing up blood. | 55 (58.5) | 39 (41.5) | | | |
| 5. Asks DM patients about noticeable weight loss. | 8 (8.5) | 86 (91.5) | | | |
| 6. Provides TB education to DM patients. | 26 (27.7) | 68 (72.3) | | | |
| 7. Provides counseling on TB screening to DM patients. | 26 (27.7) | 68 (72.3) | | | |
| 8. Refers DM patients for TB screening. | 6 (6.4) | 88 (93.6) | | | |
| | Never n (%) | Rarely n (%) | For All New Clients n (%) | For most clients n (%) | All the Time for All Clients n (%) |
| 9. How often do you ask DM patients about a cough lasting two weeks or more? | 58 (61.7) | 2 (2.1) | 11 (11.7) | 1 (1.1) | 22 (23.4) |
| 10. How often do you ask DM patients about excessive night sweats of any duration? | 52 (55.3) | 13 (13.8) | 6 (6.4) | 2 (2.1) | 21 (22.4) |
| 11. How often do you ask DM patients if they had a fever? | 5 (5.3) | 3 (3.2) | 9 (9.6) | 4 (4.3) | 73 (77.6) |
| 12. How often do you ask DM patients about chest pain or coughing up blood? | 55 (58.5) | 4 (4.3) | 5 (5.3) | 6 (6.4) | 24 (25.5) |
| 13. How often do you check for noticeable weight loss (≥3 kg in a month) among DM patients? | 8 (8.5) | 16 (17.0) | 10 (10.6) | 12 (12.8) | 48 (51.1) |
| 14. How often do you provide TB education to patients with DM? | 26 (27.7) | 1 (1.1) | 65 (69.2) | 1 (1.1) | 1 (1.1) |
| 15. How often do you provide counseling on TB screening to patients with DM? | 26 (27.7) | 12 (12.8) | 18 (19.2) | 4 (4.3) | 34 (36.0) |
| 16. How often do you refer DM patients with presumptive TB signs to a TB clinic for diagnosis and treatment? | 6 (6.4) | 3 (3.2) | 10 (10.6) | 10 (10.6) | 65 (69.2) |

The overall fidelity score of provider-level IF on the adherence to TB screening among providers delivering DM services according to the National TB Guideline, as shown in Table 5. The Overall fidelity score was derived from three domain components: TB symptoms/signs, TB education and counseling, and TB referral across the two constructs of content and frequency.

**Table 5. Overall provider -level implementation fidelity of TB screening for DM patients (n = 94).**

| Component | Low fidelity n (%) | High fidelity n (%) |
|---|---|---|
| TB signs/symptoms | 7 (7.4) | 87 (92.6) |
| TB awareness (education & counseling) awareness | 27 (28.7) | 67 (71.3) |
| TB referrals | 6 (6.4) | 88 (93.6) |
| Over all | 16 (17.0) | 78 (83.0) |

### TB signs and symptoms

Out of 94 DM providers, 92.6% (n = 87) had scores above the cut-off point score on asking about the TB signs/symptoms for DM patients.

### TB education and counseling

Moreover, on TB education and counselling among DM providers, out of 94 DM providers, 71.3% (n = 67) had scores above the cut-off point on providing the TB education and counselling for DM patients.

### TB referrals

Lastly, provision of TB referrals for DM patients suspected of TB among the DM providers showed that out of 94 DM providers, 93.6% (n = 88) had scores above the cut-off point on providing TB referrals for DM patients suspected of TB based on signs and symptoms.

Thus, the overall provider -level IF score of TB screening for DM patients was higher (83.0%, n = 78) out of the 94 providers' self-reported adherence to TB screening guideline components, reflecting a robust provider-level IF, with only (17.0%, n = 16) having a lower fidelity.

Moreover, the provider-level IF of TB screening according to the National TB guideline differed across various health-care facilities, as illustrated in (S1 Fig). Hospitals and health centers showed the highest fidelity levels, with hospitals having 45.8%, and health centers 43.8%, reflecting strong adherence to the guidelines. In contrast, dispensaries recorded the lowest fidelity levels at 10.4%, indicating possible challenges in sustaining guideline compliance within these facilities.

### Provider-level implementation fidelity of TB screening among DM patients based on social-demographic characteristics

The provider-level IF, based on their social -demographic characteristics as shown in Table 6. The findings showed that female staff (n = 24;50%,p = 0.534), providers aged 18–34 years (n = 29;60.42%,p = 0.341), providers with diploma and below level of education (n = 28;58.83%,p = 0.860), providers with one to five years of working experience (n = 36;75.00%,p = 0.902) and those working in hospitals (n = 22;45.83%,p = 0.034).

Overall, the findings suggest that the provider-level IF was significantly associated with the health facility type in which the provider was working.

### Social-demographic factors associated with provider-level implementation fidelity of TB screening among DM patients

We used the modified Poisson regression on social demographics characteristics to identify factors associated with the provider-level IF of TB screening among DM patients, as indicated in Table 7. Among the social demographics of study participants, the level of the facility was strongly associated with higher provider-level IF, as DM providers at hospitals

PLOS Global Public Health

**Table 6. Provider-level implementation fidelity based on social-demographic characteristics.**

| Characteristics | Low fidelity level n (%) | High fidelity level n (%) | P value |
|---|---|---|---|
| **Age categories** | | | |
| 18-34 | 32 (69.57) | 29 (60.42) | |
| 35-60 | 14 (30.43) | 19 (39.58) | 0.341 |
| **Gender** | | | |
| Male | 20 (43.48) | 24 (50.00) | |
| Female | 26 (56.52) | 24 (50.00)) | 0.534 |
| **Education level** | | | |
| Degree and above | 20 (43.48) | 20 (41.67) | |
| Diploma and below | 26 (56.52) | 28 (58.33) | 0.860 |
| **Level of facility** | | | |
| Dispensary | 15 (32.61) | 5 (10.42) | |
| Health Center | 20 (43.48) | 21 (43.75) | |
| Hospital | 11 (23.91) | 22 (45.83) | 0.034 |
| **Years of working** | | | |
| 1 to 5 years | 35 (76.09) | 36 (75.00) | |
| Above 5 years | 11 (23.91) | 12 (25.00) | 0.902 |

were three times more likely (aPR = 3.60, 95% CI 1.52-8.5; p = 0.004) to adhere to TB screening for DM patients compared to those at health centers and dispensaries. Although a provider working at a health center showed associations in the crude model, it did not retain statistical significance in the adjusted analysis.

**System, provider, and guideline -related factors associated with provider-level implementation fidelity of TB screening among DM patients**

The study found several factors to be associated with provider-level IF. Among the system-related factors, teamwork (aPR = 2.28; 95% CI: 1.11-7.12; p-value = 0.032) was significantly associated with higher provider-level fidelity as shown in Table 8. This suggests that DM health providers with strong teamwork were twice as likely to adhere to TB screening compared to those with poor teamwork. While training showed associations in the crude model, it did not retain statistical significance in the adjusted analysis. Furthermore, among the provider-related factors, self-efficacy was significantly associated with provider-level IF. This indicates that providers with self-efficacy were twice as likely to adhere to TB screening protocols for DM patients compared to those with low self-efficacy (aPR = 2.29; 95% CI: 1.04-5.02; p = 0.024). As for the case of guideline-related factors, the study found that in both bivariate and multivariate modified Poisson regression, the analysis indicated that there was no statistical association between the guideline factors and the provider-level IF. In multivariate analysis, nature and source of TB guidelines (aPR = 1.79; 95%CI:0.56-2.55; p = 0.781); relative advantage (aPR = 1.55; 95% CI: 0.64-3.75; p = 0.332) and design quality and package (aPR = 1.43; 95% CI: 0.67-3.07; p = 0.483). Although all the guideline-related factors in multivariate analysis showed an increased prevalence ratio in provider-level IF of TB screening among DM patients, the results were not statistically significant.

## Discussion

The study assessed the provider-level IF and factors associated with TB screening among DM patients in Ubungo District at public health facilities in Dar-es-salaam, Tanzania. Overall, provider-level IF of TB screening among DM patients was high, with 83.0% of providers self-reporting to adhere to guideline components. This suggests that when the National

**Table 7. Modified Poisson regression of social-demographic factors associated with provider-level implementation fidelity of TB screening among DM patients at public health facilities in Ubungo district, Dar-es-salaam region.**

| Variable | Bivariable Poisson (robust SE) | | Multivariable Poisson (robust SE) | |
|---|---|---|---|---|
| | cPR (95%CI) | p-value | Adjusted aPR (95%CI) | p-value |
| **Age** | | | | |
| 18-34 | Ref | | | |
| 35-60 | 1.21 (0.81-1.79) | 0.342 | 1.39 (0.72-1.63) | 0.684 |
| **Sex** | | | | |
| Female | Ref | | | |
| Male | 0.88 (0.59-1.30) | 0.484 | 1.86 (0.58-2.23) | 0.492 |
| **Education level** | | | | |
| Degree and above | Ref | | | |
| Diploma and below | 0.97 (0.64-1.44) | 0.531 | 2.21 (0.85–2.90) | 0.123 |
| **Level of facility** | | | | |
| Dispensary | Ref | | | |
| Health Center | 0.20 (0.12-0.34) | <0.001 | 2.41 (0.87-3.27) | 0.072 |
| Hospital | 1.51 (1.13-2.02) | 0.033 | 3.60 (1.52-8.50) | 0.004 |
| **Years of working** | | | | |
| 1 to 5 years | Ref | | | |
| Above 5 years | 0.97 (0.62-1.53) | 0.903 | 0.42 (0.12- 1.44) | 0.161 |

SE (Standard Error), cPR (Crude Prevalence Ratio), 95% CI (95% Confidence Interval), aPR (Adjusted Prevalence Ratio), and Ref (Reference category).

TB guideline is institutionalized and supported within routine service delivery platforms, substantial adherence can be achieved at the provider level. The findings in our study are in agreement with a study in the Republic of the Marshall Islands, which reported high provider-level IF of TB screening among DM patients [30]. However, provider-level IF was not uniform across domains. The TB education and counseling domain showed the lowest performance compared to screening symptoms and providing referrals. Similar domain-specific gaps have been reported in other TB–DM integration studies [21,31,32]. The comparatively lower performance in the education and counseling domain suggests that these activities may be more challenging to sustain consistently within routine clinical workflows. While symptom inquiry and referral processes can be embedded into standard consultation checklists, education and counseling components may require additional time, communication effort, and patient engagement. This difference in workflow integration may account for the domain-specific variation observed. Facility-level differences in provider-level IF were also evident. Hospitals demonstrated significantly higher fidelity compared to health centers and dispensaries. Furthermore, other facility-level variations in provider-level IF have reported similar findings in guideline adherence that have been documented in studies from Tanzania and other low- and middle-income countries [33,34]. The higher provider-level IF observed in hospitals may reflect structural and organizational characteristics typically associated with higher-level facilities. Hospitals often operate within more formalized administrative systems and may benefit from stronger service integration mechanisms, which can support consistent adherence to clinical guidelines. In contrast, the lower fidelity observed in dispensaries may reflect systemic differences inherent within decentralized primary care settings. Lower-level facilities frequently operate in resource-constrained environments, where competing clinical demands and operational limitations can affect the consistency of guideline implementation. These contextual dynamics may contribute to the disparities identified across facility types.

The study also examined the factors associated with provider-level IF of TB screening among DM patients. The findings showed that among the system characteristics, teamwork was significantly associated with higher provider IF of TB

**Table 8. Modified Poisson regression of system, provider, and guideline-related factors associated with provider-level implementation fidelity of TB screening among DM patients at public health facilities in Ubungo district, Dar-es salaam region.**

| Variable | Bivariable Poisson (robust SE) | | Multivariable Poisson (robust SE) | |
|---|---|---|---|---|
| | cPR (95%CI) | p-value | Adjusted aPR (95%CI) | p-value |
| **System factors** | | | | |
| Teamwork | 2.08 (1.23-4.97) | <0.010 | 2.28 (1.11-7.12) | 0.032 |
| Training | 0.62 (0.40-0.94) | 0.021 | 0.63 (0.38-1.09) | 0.103 |
| staffing | 0.91 (0.49- 1.70) | 0.772 | 0.76 (0.33-1.77) | 0.534 |
| Records&Documentation | 1.12 (0.60-2.10) | 0.364 | 1.03 (0.43- 2.48) | 0.952 |
| **Provider factors** | | | | |
| Self-efficacy | 2.40 (1.22-4.67) | <0.010 | 2.29 (1.04-5.02) | 0.024 |
| Familiarity to TB guideline | 1.66 (1.13- 2.43) | <0.010 | 1.19 (0.73- 1.93) | 0.483 |
| **Guideline factors** | | | | |
| Nature and source of TB guidelines | 1.16 (0.54- 2.50) | 0.694 | 1.79 (0.56- 2.55) | 0.781 |
| Relative advantage | 1.01 (0.52- 1.95) | 0.973 | 1.55 (0.64- 3.75) | 0.332 |
| Design quality and package | 1.40 (0.72- 2.75) | 0.324 | 1.43 (0.67- 3.07) | 0.483 |

SE (Standard Error), cPR (Crude Prevalence Ratio), 95% CI (95% Confidence Interval), aPR (Adjusted Prevalence Ratio).

screening among DM patients. These findings align with evidence from a systematic review study that highlights team-work as a critical determinant of guideline implementation in healthcare settings [35]. The significant association between teamwork and higher provider-level IF underscores the potential importance of collaborative practice environments in strengthening guideline adherence. Team-based coordination may facilitate clearer role allocation, shared accountability, and structured screening routines, thereby promoting more consistent implementation of TB screening activities within DM clinics. Moreover, self-efficacy was also significantly associated with higher fidelity of TB screening for DM patients. These findings are consistent with previous studies that demonstrated that higher self-efficacy among healthcare providers was associated with improved adherence to clinical guidelines and evidence-based practices [36,37]. This may suggest that higher perceived capability may support consistent execution of screening procedures within routine care delivery. Therefore, these findings suggest that high provider-level IF of TB screening among DM patients is achievable within routine public health services in an urban low-resource setting. However, the observed disparities across facility levels highlight persistent structural inequities within decentralized health systems. Therefore, these findings suggest that provider-level implementation fidelity of TB screening among patients with DM can be achieved within routine public health services. However, the observed variability across facility levels and domains highlights potential structural differences in implementation contexts. Therefore, factors such as teamwork and provider self-efficacy may play an important role in shaping implementation fidelity. While these findings provide useful insights into potential areas for strengthening implementation, further research is needed to determine effective strategies, particularly in lower-level health facilities.

## Limitations

Although this study provides valuable insights into provider-level IF on TB screening among DM patients at public health facilities in Ubungo District, the following limitations should be considered when interpreting these findings. First, the assessment relied on provider self-reported data, which introduces the possibility of social desirability, which may overestimate actual implementation practices [38–40]. Second, the absence of observational or record-based verification limits the objective assessment of the actual practice [41]. Third, clustering at the facility level was not explicitly accounted for in the analysis, which may have introduced intra-cluster correlation and affected the precision of the estimates. Lastly, the findings may have limited generalizability to private facilities.

## Conclusion

Provider-level implementation fidelity of TB screening among patients with DM was high and significantly associated with teamwork, provider self-efficacy, and facility level. However, variability across domains and facility types indicates inconsistencies in implementation. These findings highlight the importance of provider-level and facility-level factors in shaping implementation fidelity within routine diabetes care.

## Supporting information

**S1 Fig. Provider-level implementation fidelity of TB screening among DM patients by facility level.** This figure shows the distribution of provider-level implementation fidelity scores for Tuberculosis (TB) screening among Diabetes mellitus (DM) patients, stratified by health facility levels (hospitals, health centers, and dispensaries) in Ubungo District, Dar es Salaam, Tanzania.
(TIF)

**S1 Text. Questionnaire on provider-level implementation fidelity of TB screening for DM patients.** This document contains the full structured questionnaire used to assess provider-level implementation fidelity, based on content and frequency details. The questions are grouped into three domains: screening for TB signs and symptoms, patient education and counseling, and referrals, in accordance with the National TB and Leprosy Program guidelines.
(DOCX)

## Acknowledgments

The authors wish to sincerely thank everyone who contributed to making this study possible, including the Ubungo district authorities. We are also grateful to all study participants for their cooperation, which greatly contributed to the successful completion of this research. This study was undertaken as part of a Master's degree in Project Management, Monitoring, and Evaluation in Health at Muhimbili University of Health and Allied Sciences.

## Author contributions

**Conceptualization:** Edwin Christian Chavala.

**Data curation:** Edwin Christian Chavala.

**Formal analysis:** Edwin Christian Chavala.

**Funding acquisition:** Edwin Christian Chavala.

**Investigation:** Edwin Christian Chavala.

**Methodology:** Edwin Christian Chavala.

**Project administration:** Edwin Christian Chavala.

**Resources:** Edwin Christian Chavala.

**Software:** Edwin Christian Chavala.

**Supervision:** Edwin Christian Chavala.

**Validation:** Linda Simon Paulo, Tumaini Nyamhanga.

**Visualization:** Edwin Christian Chavala.

**Writing – original draft:** Edwin Christian Chavala.

**Writing – review & editing:** Felistar William Mwakasungura, Linda Simon Paulo, Tumaini Nyamhanga.

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
