## [Decision Letter · Decision Letter 0]

18 Jan 2026

PGPH-D-25-03950

Implementation Fidelity of Tuberculosis Screening for Diabetes Mellitus Patients among healthcare providers offering Diabetes services in Ubungo District, Dar es Salaam, Tanzania

Dear Dr. Chavala,

Thank you for submitting your manuscript to PLOS Global Public Health. After careful consideration, we feel that it has merit but does not fully meet PLOS Global Public Health’s publication criteria as it currently stands. Therefore, we invite you to submit a revised version of the manuscript that addresses the points raised during the review process.

We look forward to receiving your revised manuscript.

Kind regards,

Joel Msafiri Francis, MD, MS, PhD

Academic Editor

Journal Requirements:

1. Please ensure that your Ethics Statement is available in its entirety at the beginning of your Methods section, under a subheading 'Ethics Statement'.

2. Please ensure that the Title in your manuscript file and the Title provided in your online submission form are the same.

4. For studies involving third-party data, we encourage authors to share any data specific to their analyses that they can legally distribute. PLOS recognizes, however, that authors may be using third-party data they do not have the rights to share. When third-party data cannot be publicly shared, authors must provide all information necessary for interested researchers to apply to gain access to the data. (https://journals.plos.org/plosone/s/data-availability#loc-acceptable-data-access-restrictions

Additional Editor Comments (if provided):

Reviewers' comments:

Reviewer's Responses to Questions

**Comments to the Author**

1. Does this manuscript meet PLOS Global Public Health’s publication criteria? Is the manuscript technically sound, and do the data support the conclusions? The manuscript must describe methodologically and ethically rigorous research with conclusions that are appropriately drawn based on the data presented.? Is the manuscript technically sound, and do the data support the conclusions? The manuscript must describe methodologically and ethically rigorous research with conclusions that are appropriately drawn based on the data presented.

Reviewer #1: Yes

Reviewer #2: Partly

2. Has the statistical analysis been performed appropriately and rigorously?

Reviewer #1: Yes

Reviewer #2: No

3. Have the authors made all data underlying the findings in their manuscript fully available (please refer to the Data Availability Statement at the start of the manuscript PDF file)?

The PLOS Data policy requires authors to make all data underlying the findings described in their manuscript fully available without restriction, with rare exception. The data should be provided as part of the manuscript or its supporting information, or deposited to a public repository. For example, in addition to summary statistics, the data points behind means, medians and variance measures should be available. If there are restrictions on publicly sharing data—e.g. participant privacy or use of data from a third party—those must be specified.requires authors to make all data underlying the findings described in their manuscript fully available without restriction, with rare exception. The data should be provided as part of the manuscript or its supporting information, or deposited to a public repository. For example, in addition to summary statistics, the data points behind means, medians and variance measures should be available. If there are restrictions on publicly sharing data—e.g. participant privacy or use of data from a third party—those must be specified.

Reviewer #1: Yes

Reviewer #2: Yes

4. Is the manuscript presented in an intelligible fashion and written in standard English?

Reviewer #1: Yes

Reviewer #2: Yes

5. Review Comments to the Author

Reviewer #1: This manuscript addresses a highly relevant public health issue. Overall the manuscript is well-structured and presents important findings however, a few refinements could enhance clarity. Specifically, the discussion could be strengthened by drawing clearer implications for policy and scalability,how lessons from high-fidelity can be adapted to low fidelity settings. Adding explanatory footnotes to some of the tables and ensuring that figures and tables, supporting materials are properly referenced intext .

Reviewer #2: This manuscript addresses an important and understudied implementation science topic: implementation fidelity of tuberculosis (TB) screening among diabetes mellitus (DM) patients in routine care settings in Tanzania. The topic is relevant to TB–DM collaborative activities and aligns well with global priorities. However, several substantive issues need to be addressed before the findings can be interpreted with confidence.

Sampling strategy

There is a lack of clarity and internal consistency between the sampling strategy described in the Methods and the way provider numbers are reported in the Results. The Methods indicate that 2–4 healthcare providers were selected per facility using proportional allocation and simple random sampling, yet Table 1 reports aggregate numbers by facility type (e.g., dispensary, health centre, hospital) without indicating how many providers were recruited from each of the 20 facilities. This makes it difficult to assess representativeness and raises concerns about clustering (e.g., whether multiple providers came from the same facility). The authors should clearly report the number of participants recruited per facility, ideally in a supplementary table, and explain how the stated sampling strategy was operationalised.

Outcome definition and interpretation

The primary outcome is provider-level implementation fidelity, measured through self-reported adherence to TB screening guideline components. However, the Results and Discussion repeatedly imply patient-level screening coverage (e.g., statements suggesting that a certain proportion of DM patients were screened for TB). No patient-level numerator or denominator is presented, and the Methods do not describe record review or observation. The authors should consistently frame the outcome as provider-level fidelity, revise language that implies patient screening coverage, and explicitly acknowledge the absence of patient-level screening data as a limitation if such data were not collected.

Unsupported causal explanations

The Discussion attributes low implementation fidelity (17%) to factors such as lack of integrated TB–DM training and provider role allocation, yet these explanations are not adequately supported by the study data. Training does not appear to remain significant in adjusted analyses, and several explanatory statements are not referenced. In addition, the Discussion suggests that degree-holding providers may focus on administrative duties, while the Methods state that staff in administrative roles were excluded from the study. These contradictions should be resolved, and causal language should be softened or removed where not directly supported by evidence.

Discussion focus

The Discussion begins by restating the study’s aim and strengths rather than clearly summarising the key findings. Several paragraphs repeat results or focus heavily on comparisons with other studies, with limited interpretation of what the findings mean for the Ubungo or Tanzanian primary care context. The Discussion would be strengthened by focusing on (i) the most poorly implemented screening components, (ii) why dispensaries showed lower fidelity, and (iii) the implications for TB–DM integration, supervision, and training in similar settings.

Statistical reporting

The analytical approach (modified Poisson regression) is appropriate for the outcome, but there appear to be potential reporting errors (e.g., confidence intervals in Table 4 where bounds appear inconsistent). These should be carefully checked. In addition, typographical errors (e.g., “modified poison regression”) should be corrected.

Limitations section

The limitations are acknowledged; however, they could be more clearly framed from an implementation science perspective, including reliance on self-reported practices, absence of observational or record-based verification, and the cross-sectional design limiting causal inference.

6. PLOS authors have the option to publish the peer review history of their article (what does this mean?). If published, this will include your full peer review and any attached files.). If published, this will include your full peer review and any attached files.

**Do you want your identity to be public for this peer review?** For information about this choice, including consent withdrawal, please see our Privacy Policy..

Reviewer #1: No

Reviewer #2: No

Figure Resubmissions:

---

## [Decision Letter · Decision Letter 1]

18 Mar 2026

PGPH-D-25-03950R1

Implementation Fidelity of Tuberculosis Screening for Diabetes Mellitus Patients Among Healthcare Providers Offering Diabetes Services in Ubungo District, Dar-es-salaam, Tanzania.

Dear Dr. Chavala,

Thank you for submitting your manuscript to PLOS Global Public Health. After careful consideration, we feel that it has merit but does not fully meet PLOS Global Public Health’s publication criteria as it currently stands. Therefore, we invite you to submit a revised version of the manuscript that addresses the points raised during the review process.

We look forward to receiving your revised manuscript.

Kind regards,

Joel Msafiri Francis, MD, MS, PhD

Academic Editor

Journal Requirements:

Additional Editor Comments (if provided):

Reviewers' comments:

Reviewer's Responses to Questions

**Comments to the Author**

1. If the authors have adequately addressed your comments raised in a previous round of review and you feel that this manuscript is now acceptable for publication, you may indicate that here to bypass the “Comments to the Author” section, enter your conflict of interest statement in the “Confidential to Editor” section, and submit your "Accept" recommendation.

Reviewer #1: All comments have been addressed

Reviewer #2: (No Response)

2. Does this manuscript meet PLOS Global Public Health’s publication criteria? Is the manuscript technically sound, and do the data support the conclusions? The manuscript must describe methodologically and ethically rigorous research with conclusions that are appropriately drawn based on the data presented.? Is the manuscript technically sound, and do the data support the conclusions? The manuscript must describe methodologically and ethically rigorous research with conclusions that are appropriately drawn based on the data presented.

Reviewer #1: Yes

Reviewer #2: Partly

3. Has the statistical analysis been performed appropriately and rigorously?

Reviewer #1: Yes

Reviewer #2: No

4. Have the authors made all data underlying the findings in their manuscript fully available (please refer to the Data Availability Statement at the start of the manuscript PDF file)?

The PLOS Data policy requires authors to make all data underlying the findings described in their manuscript fully available without restriction, with rare exception. The data should be provided as part of the manuscript or its supporting information, or deposited to a public repository. For example, in addition to summary statistics, the data points behind means, medians and variance measures should be available. If there are restrictions on publicly sharing data—e.g. participant privacy or use of data from a third party—those must be specified.requires authors to make all data underlying the findings described in their manuscript fully available without restriction, with rare exception. The data should be provided as part of the manuscript or its supporting information, or deposited to a public repository. For example, in addition to summary statistics, the data points behind means, medians and variance measures should be available. If there are restrictions on publicly sharing data—e.g. participant privacy or use of data from a third party—those must be specified.

Reviewer #1: Yes

Reviewer #2: No

5. Is the manuscript presented in an intelligible fashion and written in standard English?

Reviewer #1: Yes

Reviewer #2: Yes

6. Review Comments to the Author

Reviewer #1: (No Response)

Reviewer #2: Thank you for the revision. The manuscript has improved, especially in its clearer framing of provider-level implementation fidelity and correction of the “modified Poisson regression” terminology. However, several issues still need attention before the manuscript is fully satisfactory.

First, although the reporting of sampling has improved, the manuscript should still clarify whether clustering by facility was considered analytically or acknowledge this as a limitation, since providers were sampled from 20 facilities and multiple providers may have come from the same facility. This remains relevant to interpretation of the estimates.

Second, the outcome is now framed more consistently as provider-level implementation fidelity, which is appreciated. However, the manuscript should still be checked carefully to ensure that no wording implies patient-level screening coverage, because the study was based on provider self-report rather than patient-level numerator/denominator data, record review, or observation. The limitations section also still needs to state this more explicitly.

Third, the conclusions remain somewhat stronger than the data directly support. The manuscript states that gaps observed in education/counseling and lower fidelity in dispensaries underscore the need for integrated TB-DM training and supervisory visits, yet these recommendations go beyond what was directly demonstrated in the adjusted analyses. These points may be reasonable implementation implications, but they should be framed more cautiously.

Fourth, the regression tables still need careful verification. Although the response states that all confidence intervals in Tables 7 and 8 were corrected, there is still at least one clear internal inconsistency: in Table 8, “Design quality” is reported with an adjusted PR of 1.43 and a 95% CI of 0.67–1.37, which is not possible because the point estimate lies outside the confidence interval. This indicates that the tables should be rechecked thoroughly before acceptance.

Finally, the references require a careful audit for appropriateness and reliability. The manuscript cites a Google search URL as reference 2 for the WHO global TB burden statement, which is not an acceptable bibliographic source for such a key claim. In addition, the limitations section cites reference 41 to support the lack of observational or record-based verification, yet reference 41 is a paper on the International Court of Justice and is clearly unrelated to implementation fidelity or health research methods. These issues raise concern that the reference list has not yet been fully checked for citation-to-claim matching, completeness, and use of authoritative sources.

Overall, the manuscript is improved, but I do not think all important concerns are fully resolved. I recommend a further minor revision focused on clarifying residual interpretation issues, softening over-extended conclusions, correcting the regression tables, and auditing the references carefully.

7. PLOS authors have the option to publish the peer review history of their article (what does this mean?). If published, this will include your full peer review and any attached files.). If published, this will include your full peer review and any attached files.

**Do you want your identity to be public for this peer review?** For information about this choice, including consent withdrawal, please see our Privacy Policy..

Reviewer #1: **Yes:** Cresensia felician muhereCresensia felician muhere

Reviewer #2: No

 Figure Resubmissions:

---

## [Editor Report · Decision Letter 2]

14 Apr 2026

Implementation Fidelity of Tuberculosis Screening for Diabetes Mellitus Patients Among Healthcare Providers Offering Diabetes Services in Ubungo District, Dar-es-salaam, Tanzania

PGPH-D-25-03950R2

Dear Mr Chavala,

We are pleased to inform you that your manuscript 'Implementation Fidelity of Tuberculosis Screening for Diabetes Mellitus Patients Among Healthcare Providers Offering Diabetes Services in Ubungo District, Dar-es-salaam, Tanzania' has been provisionally accepted for publication in PLOS Global Public Health.

Best regards,

Joel Msafiri Francis, MD, MS, PhD

Academic Editor